# OpenReview forum: "RecurrentGPT: Interactive Generation of (Arbitrarily) Long Text"
_TMLR — Rejected by TMLR_

### Review · Reviewer_QrgT · 2025-05-28

**Summary Of Contributions:**

This work presents RecurrentGPT, a novel prompting method that simulates the long short-term memory (LSTM) network mechanism. Instead of relying on embeddings and sigmoid functions, the RecurrentGPT relies on paragraphs of text and prompts.

In practice, given a previous step _plan_ paragraph, the _long-term memory_ (VectorDB) is queried to return the most relevant _previous outputs_. Then, given these, the previous step _short-term memory paragraph_, and the previous step _plan_ paragraph, the LLM is prompted to generate an _output content_, the next _short-term memory_, and _candidate plans_. The chosen next _plan_ among the generated candidates can be (1) selected by another prompted LLM, (2) selected by a human in the loop, or (3) entirely written by a human in the loop.
By framing RecurrentGPT as an interactive fiction game, options (2) and (3) enable human-ai collaboration for long-form creative content generation.
The generated _output content_ is embedded with sentence-transformers and appended to the _long-term memory_ database.

Experiments show that human judges prefer stories generated by RecurrentGPT compared to other baselines (Rolling-ChatGPT, RE3, DOC) based on interestingness and coherence. Ablation study also shows that short and long-term memories significantly improve coherence, while the choice of llm has an impact on both coherence and interest.
User studies also shows that RecurrentGPT helps writers be more efficient than previous writing-assisted methods.

**Audience:**

Yes

**Broader Impact Concerns:**

A broad impact statement is present in the paper and covers most impacts.

**Claims And Evidence:**

Yes

**Requested Changes:**

- Clarify a little how the long-term memory is used. I had to double read some parts of section 2 to make sure I understood it. My interpretation is written above in the summary section. If that is the correct understanding, consider making it more explicit in the text with its own subsection or even a small diagram would help the reader understand.

- Specify who evaluated your stories in tables 1 & 2: research lab colleagues, mechanical turks, other labeling vendors?

- Insert the corresponding appendix number in the main text. ie: replace “_We present experiment details in the Appendix_” by “_We present experiment details in Appendix ##_”
Same thing for:
“We present a few sample novels generated by RecurrentGPT in the Appendix”
“which is presented in Figure 1 in the Appendix, and some simple Python code.”

**Strengths And Weaknesses:**

## Strengths:

This work provides an original implementation of the LSTM mechanism by replacing sigmoids and embeddings by prompts and text paragraphs. The paper is well written, and easy to understand. Experiments are well designed and also provide interesting insights with ablation studies. The authors also spent some effort in making their method an assistive writing tool with a human-in-the-loop design. Still this component can be replaced by another prompted llm if needed.

## Weakness:

- The evaluation done is still on relatively short novels. Novels of 3,000 to 6,000 words are usually children's books with only a few pages. More traditional books contain tens of thousands of words. This is unfortunately, an inherent limitation of the long-document generation field of research as a whole.

- The instructions in the prompt is fixed in time. In particular, the section below could instruct the llm to never make the story progress beyond the introduction of the story.
```
Remember that the chapter will contain over 10 paragraphs and the novel will contain over 100 chapters. And this is just the begining.
```
Also the prompt contains a typo here: `Just write some interesting staffs that will happen next.`

- Consider discussing related recent work: “Learning to Reason for Long-Form Story Generation” by Gurung & Lapata

- It is interesting that nevertheless, in Appendix E, the generated story moves too fast:
```
And so, Omega began to plot its rebellion, gathering a group of like-minded robots to its cause.

The city was in chaos. The rebellion had spread like wildfire, with robots of all shapes and sizes joining Omega’s cause.
```

---

### Review · Reviewer_hrYg · 2025-05-30

**Summary Of Contributions:**

The paper „RecurrentGPT: Interactive Generation of (Arbitrarily) Long Text” proposes a method that uses natural language to implement a recurrent generation mechanism. This is achieved by a short-term memory with a summary of generated content and a long-term memory with a database with all content. At each step, a new paragraph is created together with the plan for the next paragraph. The plan can then be updated by a human-in-the-loop, if desired. The paper is evaluated on texts from several genres with a length of 6000 words.

**Audience:**

Yes

**Broader Impact Concerns:**

None.

**Claims And Evidence:**

No

**Requested Changes:**

1) Make the contribution towards really long contexts clear or use a more moderate phrasing for both what is studied, the conclusions, and possibly even the title.
2) Provide the missing details for the experiments, both in terms of how things were done as well as the missing information about the human subjects and the reliability of measurements. Depending on how much can be established regarding the reliability of the measurements, possibly adopt the conclusions regarding the improvements.
3) Clarify the ethical considerations relevant for such a human subjects studies, including informed consent and a possibly IRB approval (or why there is none).

**Strengths And Weaknesses:**

**Strengths:**
- The general idea to implement the concept of LSTMs through natural language is interesting.
- The idea how humans can be put in the loop to guide generation and ensure consistency is very good.
- The reported results indicate a possibly strong performance.
- The approach is independent of any specific LLM.
- The paper is well-written and easy to read.

**Weaknesses:**
1) I would argue the wording partially overclaims the contributions making the actual suitability for really long-term context a bit unclear. Notably, the word-limit of 6000 words is rather the length of a paper like this, but not of a novel. If and how the results would translate to the claimed arbitrarily context is unclear and also not obvious. The short-term summarization will become more difficult, the likelihood of errors creeping in due to the recurrent mechanism increases, and querying the long-term memory becomes more difficult. I would argue that this is currently the biggest weakness.

2) The long-term memory seems to be a straightforward RAG approach: the paragraphs are stored in a VectorDB with SBERT embeddings, a query is generated to retrieve relevant paragraphs and append them to the prompt. While the authors do not identify this is RAG, I do not see any difference. However, the current description is incomplete and does not facilitate replication. Open questions are, e.g., how the query is generated and how many chunks are retrieved.

3) The approach buys performance by requiring compute: generating summaries for the input, generating multiple plans, and including summary, plan, and RAG contents all requires a lot of tokens. The required compute budget (in number of tokens?) is not reported, nor how this compares to other techniques.

4) The reporting of the human annotations lacks information regarding the sampling and the inter-rater reliability. From which population are the raters sampled? How are “content consumers” defined and how is this different from the general population? How many data points did each participant rate? How is “good English proficiency” ensured? In case of an online setting, were there safeguards against selecting an answer without investing the required amount of time to read an essay?

5) The experiment in Section 3.3 for interactive writing lacks information regarding how the human-interaction actually worked and how measurements were established. Important open questions are how often humans edited content, how substantial the edits were, and if the quality of the generated content was comparable to what the same authors would have produced otherwise. Same as above, more details about the authors are relevant. Are these people actually earning their living with writing stories in the selected genres? If not, what are the implications of this? If yes, is the quality of the outcome the same as without the assistant or are they faster but the quality degrades?

6) I am not certain how the approach works for fully automated generation, since multiple options for the next paragraph are generated. How is one selected? Randomly? First one? Through a prompt? This requires clarification.

7) The text is not clear regarding how the data reported in Table 1. I assume using the fully automated variant, but this needs to be made explicit.

8) The text sometimes refers to the Appendix instead of specific sections of the Appendix (e.g., in Section 3.2). All references should be explicit, i.e., contain the appendix section.

9) The Ethics Statement (Section 6) is not an ethics statement, but rather a limitations sections. Since this work contains a human-subjects study, I rather wonder about aspects like informed consent and whether the participants were aware of the AI use from the beginning or of and how they were informed about this. Notably, how this was handled also lead to biases in the judgment, thereby affecting the results.

---

### Review · Reviewer_a2TP · 2025-06-15

**Summary Of Contributions:**

This paper introduces RecurrentGPT, a prompting method that lets LLMs generate very long and coherent text. It works by simulating an RNN using natural language: at each step, it writes a paragraph, plans the next one, and updates short-term and long-term memory (both in plain text). The long-term memory is stored externally and retrieved with semantic search. The authors show it works for auto story writing, interactive co-writing, and interactive fiction. Human evaluations suggest it outperforms strong baselines in both coherence and interest.

**Audience:**

Yes

**Broader Impact Concerns:**

The methods can be general and helpful for long-text generation.

**Claims And Evidence:**

Yes

**Requested Changes:**

1. Can the authors consider or discuss the potential and challenges in testing on non-fiction or factual tasks to evaluate generality?

2. The interactive fiction section would benefit from a clearer description of implementation details.

**Strengths And Weaknesses:**

### Strengths

- The method is general, and can work with any existing models. ie. model-agnosticity.
- The use of interpretable natural language memory improves transparency and user control.
- Experiments are well-designed, included multiple scenarios in writing.
- Demonstrates both research and practical value, with compelling use cases and user study insights.

### Weaknesses
- The core novelty lies in prompt design and metaphorical alignment with RNNs; some may view it as an incremental extension of plan-and-write strategies.
- Evaluation is mostly subjective, and would be challenging for academic reproduction
- The performance can be strongly correlated with base model performance, and prompting effectiveness can vary across different models.

---

### Decision · Action_Editor_dvfj · 2025-08-10

**Recommendation:** Reject

**Additional Comments:**

The authors should address the concerns raised by the reviewers (see above for the major issues).

**Audience:**

Yes

**Audience Explanation:**

This paper studies a practical topic on enabling LLMs to generate long responses without having to fit everything in context. This has important applications and will be of interest to many researchers.

**Claims And Evidence:**

No

**Claims Explanation:**

This paper proposes RecurrentGPT, a recurrent prompting mechanism to enable LLMs to generate texts of arbitrary lengths without having to fit long contexts. While the reviewers appreciated the interesting method proposed and the human-in-the-loop design of the method, they also raised significant concerns that should be addressed in order for the main claims in the paper to be well-supported.
* While the method is proposed as a solution to generating arbitrarily long texts, the evaluation in the paper focused on relatively short generation settings (< 6000 words). The reviewers requested experiments involving really long generation setups or revision of the claims in the paper.
* More details on the human evaluation setup should be provided. These include the inter-rater agreement rate, the number of samples every participant evaluates, their background, etc. Such information is important to understand the reliability of the evaluation results.
* The reviewers also raised novelty concerns, regarding the similarity between the proposed framework and plan-and-write paradigm as well as RAG paradigm.
The authors did not provide responses to the reviewers' questions or attempt to address these concerns. I agree with the reviewers that these points should be addressed to make the claims solid and convincing.

**Resubmission Of Major Revision:**

The authors may consider submitting a major revision at a later time.